# Deep Fusion of Skeleton Spatial–Temporal and Dynamic Information for Action Recognition

**DOI:** 10.3390/s24237609

**Published:** 2024-11-28

**Authors:** Song Gao, Dingzhuo Zhang, Zhaoming Tang, Hongyan Wang

**Affiliations:** 1Aviation Maintenance NCO Academy, Air Force Engineering University, Xinyang 464007, China; gaosong1858@163.com (S.G.); zhaomingtang1984@163.com (Z.T.); 2College of Information Engineering, Dalian University, Dalian 116622, China; zhangdingzhuo@s.dlu.edu.cn; 3School of Comuputer Science and Technology, Zhejiang Sci-Tech University, Hangzhou 310018, China

**Keywords:** skeleton spatial–temporal feature, skeleton dynamic feature, feature enhancement, action recognition, two stream convolutional neural networks

## Abstract

Focusing on the issue of the low recognition rates achieved by traditional deep-information-based action recognition algorithms, an action recognition approach was developed based on skeleton spatial–temporal and dynamic features combined with a two-stream convolutional neural network (TS-CNN). Firstly, the skeleton’s three-dimensional coordinate system was transformed to obtain coordinate information related to relative joint positions. Subsequently, this relevant joint information was encoded as a color texture map to construct the spatial–temporal feature descriptor of the skeleton. Furthermore, physical structure constraints of the human body were considered to enhance class differences. Additionally, the speed information for each joint was estimated and encoded as a color texture map to achieve the skeleton motion feature descriptor. The resulting spatial–temporal and dynamic features were further enhanced using motion saliency and morphology operators to improve their expression ability. Finally, these enhanced skeleton spatial–temporal and dynamic features were deeply fused via TS-CNN for implementing action recognition. Numerous results from experiments conducted on the publicly available datasets NTU RGB-D, Northwestern-UCLA, and UTD-MHAD demonstrate that the recognition rates achieved via the developed approach are 86.25%, 87.37%, and 93.75%, respectively, indicating that the approach can effectively improve the accuracy of action recognition in complex environments compared to state-of-the-art algorithms.

## 1. Introduction

As a prominent research area in computer vision, human action recognition plays a crucial role in intelligent surveillance, human–computer interaction, video retrieval, and other applications. The recognition of human body actions based on RGB data has shown poor robustness due to factors such as illumination changes and cluttered backgrounds [1,2]. To address these limitations, advanced research works [3,4] have proposed the integration of depth information into human action recognition methods, aiming for higher accuracy and improved robustness against environmental changes, due to the fact that depth information is less affected by illumination changes and can effectively filter out irrelevant texture and color information from the background. However, it should be noted that the redundancy in depth image information results in increased computational complexity, which limits its practical application.

With the continuous advancement of sensor technology, affordable depth cameras such as Intel RealSense3D [5] and Xtion PRO have gained popularity. These cameras enable the easy acquisition of 3D human joint coordinates from depth maps, which contain abundant motion information. The 3D skeletal data not only provide rich depth information, but also offer advantages like simplicity of format, comprehensive motion details, and straightforward calculations. Consequently, skeletal information has gradually become the primary focus of research on human action recognition. However, effectively extracting motion information from 3D skeletal data to recognize human movements remains a significant challenge due to the noise present in the raw skeletal information captured by depth sensors and blurred spatial–temporal relationships among joints. Manual features were exploited to recognize actions in [6,7]. However, the hand-crafted feature is relatively simplistic, resulting in limited recognition accuracy and poor generalizability.

In recent years, deep learning models such as recurrent neural networks (RNNs), convolutional neural networks (CNNs), and graph convolutional networks (GCNs) have demonstrated promising results in various computer vision tasks. Building upon this progress, the field of action recognition has integrated deep learning models to improve both its generalization capabilities and its recognition performance. Leveraging the strong temporal series modeling abilities of RNNs, action recognition models have been developed in [8,9]. Nevertheless, RNNs fail to effectively capture inter-joint spatial relationships, leading to underutilization of joint space information and limited improvements in recognition performance. To address this issue, a novel hybrid spatio-temporal convolutional network was proposed in [10] to exploit the powerful spatial feature extraction capabilities of CNN for action recognition. In [11,12], action features were extracted from sequential skeleton-sequence-encoded images using CNN, where each piece of joint information was independently encoded as a color image. However, the related inter-joint information was disregarded. To overcome this limitation, a multi-task learning model was developed in [13] to capture the correlation between skeleton and action classes by maximizing potential edges. Additionally, an action recognition method based on joint distance maps (JDM) was proposed in [14,15], encoding pairwise joint distances as color images but neglecting the spatial constraints among joints during the encoding process, which leads to the confusion of joint spatial information and limited recognition accuracy. Addressing this concern, an action recognition method based on a tree skeleton diagram and reference nodes, which incorporates human structure constraints into the skeleton sequence analysis, was introduced in [16]. However, this approach solely focuses on the static characteristics of joints while ignoring their dynamic characteristics and individual participation levels during the action completion process, leading to motion information that is incompletely encoded and the loss of the spatial saliency information of joints, thereby limiting action recognition rates. It should be noted that the detection and seperation of multi-target, and the effectiveness of joint combination, as well as other issues, should be considered first in the multi-person scenario. Therefore, existing action recognition methods have mainly focused on the single-person case. Consequently, the action recognition issue for single person si considered in this paper. 

To address the aforementioned issue, this study presents two-stream CNN (TS-CNN) based action recognition method that integrates spatio-temporal and dynamic information of the skeleton. The coordinate system of the skeletal sequences is firstly transformed by the proposed method, enabling the construction of a skeleton spatial–temporal map (SSTM) that effectively captures the relative positions of joints. Furthermore, the dynamic characteristics of joints are encoded into a joint motion speed map (JMSM) to highlight differences in motion characteristics. Additionally, the SSTM and JMSM features can be enhanced by employing motion saliency and morphological operators to improve inter-class differences and reduce intra-class differences. Finally, the enhanced SSTM and JMSM can be deeply fused by using TS-CNN to effectively achieve action classification. Numerous experimental results demonstrate that the proposed skeletal spatio-temporal and dynamic feature information fusion-based recognition method outperforms state-of-the-art recognition algorithms in complex scenarios.

Along the above lines, the main contributions of this work can be summarized as follows:

(1) The coordinate system of the skeletal sequences, typically established with a depth sensor as the origin, is transformed into a body coordinate system with the hip joint serving as the origin. This transformation is justified by its relative immobility during movement, allowing for the effective representation of spatial information. By employing this formulated coordinate system, both the absolute coordinates of joints and the relative coordinates among joints are collectively encoded into a color texture map to construct SSTM characterizing the spatial–temporal features of actions.

(2) To effectively distinguish different actions implemented via joints with varying velocities, the velocity information for each joint involved in a specific action is extracted to depict the dynamic properties of the action. Consequently, the amplitude of the velocity information in each direction can be encoded as JMSM. With a view to further enhancing the representation ability of features, both spatial–temporal and dynamic features can be respectively improved by incorporating motion saliency through adjusting the color weight associated with each joint and utilizing morphology operators such as corrosion and expansion operations.

(3) To improve the performance of action recognition by effectively leveraging multiple types of information associated with skeleton sequences, TS-CNN is utilized to deeply integrate the extracted static and dynamic skeleton features, namely SSTM and JMSM. The constructed TS-CNN model comprises two enhanced versions of AlexNet, where both channels in the formulated TS-CNN possess identical structures that are meticulously designed.

The remainder of this work is organized as follows: The state-of-the-art algorithms for action recognition are depicted in Section 2. The effectiveness of the proposed approach is verified via numerical examples in Section 3. Finally, conclusions are given in Section 4.

## 2. Methods

The framework of the proposed method is illustrated in Figure 1. The proposed algorithm can be divided into the following four parts: The 3D coordinates of the skeletal sequence acquired by Kinect are first transformed to the body coordinate system with the hip joint as the origin. Subsequently, spatio-temporal and joint velocity information are encoded separately, as SSTM and JMSM. Furthermore, motion saliency is utilized to enhance joint information exhibiting significant motion characteristics in SSTM, while morphological operators are employed to improve speed information in JMSM. Finally, TS-CNN is used to extract depth information for each feature, followed by obtaining posterior probabilities based on multiplicative fusion for achieving recognition results.

### 2.1. Coordinate System Transformation

The human body can be viewed as a hinge structure consisting of the torso and limbs, where each type of movement is executed via a directional circular motion of the limb around the hip joint. However, the skeletal sequences captured by depth sensors such as Kinect are mapped to a Cartesian coordinate system with the camera as the origin (shown in Figure 2). The joint coordinates are independent of each other and fail to characterize their relative positions.

Therefore, a coordinate system transformation of the skeletal 3D coordinates is necessary to obtain a body coordinate system that effectively represents the spatial information. The coordinates of the joints in the body coordinate system are relative to the origin of the coordinates. It should be noted that the choice of the origin significantly affects the representation of the relative spatial information between the joints. The body coordinate system used in [9] takes the center of the spine as the origin, disregarding the influence of the bending motion of the upper torso on other joints. Accordingly, when the upper torso moves, the origin of the coordinates, i.e., the center of the spine, changes its position constantly, which causes the coordinate system between adjacent frames to change continually. As a result, this continual change leads to a decrease in the correlation among joints. Therefore, the joint, which is as static as possible during the movement, should be selected as the origin of coordinates to obtain relatively stable coordinate information.

With the aforementioned description, a body coordinate system with hip joint as origin because of small motion is constructed here. For a video sequence with F frames, the coordinate transformation associated with N joints can be expressed as follows:(1)$$P_{j}^{\prime f}=P_{j}^{f}-P_{*}^{f}$$
where $P_{j}^{f}$, $P_{j}^{\prime f}$ are the coordinate information of the joint j at the fth frame before and after the coordinate system transformation, and $P_{*}^{f}=(p_{*,x},p_{*,y},p_{*,z})$ is the coordinate information of the hip joint at the fth frame. The joint visualization after the transformation is shown in Figure 3.

### 2.2. Skeletal Space-Time Feature Descriptor Construction

When the body’s joints collaborate to perform specific movements, there are distinct variations in the relative positions of joints across different movements. Hence, it is crucial to extract spatial information regarding the relative positions of joints during motion to effectively characterize the movement. To effectively encode the joint spatial position information, the absolute coordinates of joints and the relative coordinates among joints are jointly encoded into a color texture map here to formulate SSTM characterizing the spatial–temporal feature of the action.

With the skeletal sequence $S=\left(P_{1}^{\prime 1},P_{2}^{\prime 2},\cdots P_{N}^{\prime F}\right)$ after coordinate transformation, the relative positions of joints can be obtained by the following equation:(2)$$C_{j\_i}^{f}=P_{i}^{\prime f}-P_{j}^{\prime f}$$
where $C_{f\_i}^{j}$ is the 3D coordinates of the jth joint in the fth frame relative to the ith joint and also depicts the spatial information of the skeleton connected to the jth and ith joint, which is the absolute coordinate of the jth joint with i=1, i.e., $C_{f\_1}^{j}=P_{j}^{\prime f}$.

With the description above, the spatial–temporal characteristics of the jth joint can be represented by the following:(3)$$Q_{j\_i}={\left[C_{j\_i}^{1},C_{j\_i}^{2},\cdots ,C_{j\_i}^{F}\right]}^{T}$$

In the skeletal structure, there are pathways among all joints, i.e., they are connected by a certain number of edges (shown in Figure 4). The fewer the edges that connect two joints, the higher the correlation between them. With this spatial constraint, only the first- and second-level correlation information (i.e., joint pairs connected by only one or two edges) is selected to reduce computational complexity and inter-class confusion, and to improve intra-class robustness. The first- and second-level correlation information is shown in the following:(4)$$R_{1}=\left[Q_{h\_k},Q_{j\_i},\cdots ,Q_{m\_n}\right],R_{2}=\left[Q_{p\_o},Q_{u\_v},\cdots ,Q_{y\_x}\right]$$
in which h, k; j, i; m, n, and so on denote joint pairs connected by only one edge; p, o, u, v, y, x, etc. illustrate joint pairs connected by two edges.

The perception region of CNN expands with network depth, necessitating the extraction of spatial information associated with joint pairs exhibiting higher correlation in shallow layers and lower correlation in deep layers. In contrast to JDM (shown in Figure 5a), proposed in [14,15], which arranges joint information as color images in a fixed order and ignores the difference of the relative spatial information, the coordinate information is arranged here according to the body structure. As depicted in Figure 4, all joints can be categorized into left-arm, right-arm, left-leg, right-leg, and torso groups arranged according to physical connections among joints. Taking the right arm as an example, the joints with numbers 9–12, 24, and 25 shown in Figure 4 are adjacent to each other, and thus have a high correlation, and they can be grouped into a parcel to extract their spatial relationship features more effectively. With the information mentioned above, the resultant SSTM can effectively encode the spatial–temporal information of joints (shown in Figure 5b).

Based on the encoded skeletal sequences, the skeletal spatio-temporal feature map can be obtained as
(5)$$E_{k}={\left[R_{1},R_{2},A\right]}^{T}={\left(\begin{array}{c} C_{25\_11}^{1}\\ C_{25\_11}^{2}\\ \vdots \\ C_{25\_11}^{F} \end{array}\begin{array}{c} \cdots \\ \cdots \\ \cdots \\ \cdots \end{array}\begin{array}{c} C_{2\_1}^{1}\\ C_{2\_1}^{1}\\ \vdots \\ C_{2\_1}^{1} \end{array}\begin{array}{c} C_{25\_10}^{1}\\ C_{25\_10}^{2}\\ \vdots \\ C_{25\_10}^{F} \end{array}\begin{array}{c} \cdots \\ \cdots \\ \cdots \\ \cdots \end{array}\begin{array}{c} C_{21\_1}^{1}\\ C_{21\_1}^{1}\\ \vdots \\ C_{21\_1}^{1} \end{array}\begin{array}{c} C_{25\_1}^{1}\\ C_{23\_1}^{2}\\ \vdots \\ C_{25\_1}^{F} \end{array}\begin{array}{c} \cdots \\ \cdots \\ \cdots \\ \cdots \end{array}\begin{array}{c} C_{2\_1}^{1}\\ C_{2\_1}^{1}\\ \vdots \\ C_{2\_1}^{1} \end{array}\right)}^{T}$$
where k is the action category, and A is the absolute coordinates of the joint points.

Let the 3D coordinates of $C_{j\_i}^{t}$ correspond to RGB channels. $E_{k}$ can be transformed as SSTM of 72×F, as shown in Figure 5b, in which one column can be reshaped as one frame, each row is associated with specific joint coordinate information, and the vertical and horizontal directions of SSTM encode the temporal and spatial information, respectively, effectively encoding the spatio-temporal information relevant to the whole action into a color texture map. Compared with the joint distances encoded in [14,15], SSTM is relatively simple to calculate and contains more abundant spatial domain information. Thus, it can effectively distinguish actions and is more robust to differences between similar actions.

### 2.3. Construction of Skeletal Motion Feature Descriptors

The completion of a specific action involves only certain joints, and different actions require the involvement of different joints with varying velocities. This fact allows for the extraction of velocity information from each joint to characterize the dynamic properties of the action. Due to individual differences in execution, there is a wide variation in velocity directions among similar actions within the same class, leading to increased divergence within intra-class dynamics. Consequently, feature descriptors that capture joint motion characteristics can be constructed by solely utilizing scalar velocity information. The velocity values of joints in the x, y, and z directions within the fth frame can be expressed as follows:(6)$$v_{x}=\left|\frac{p_{x}^{f+\Delta f}-p_{x}^{f}}{\Delta t}\right|,v_{y}=\left|\frac{p_{y}^{f+\Delta f}-p_{y}^{f}}{\Delta t}\right|,v_{z}=\left|\frac{p_{z}^{f+\Delta f}-p_{z}^{f}}{\Delta t}\right|$$
where |*| denotes the absolute value operator, $p_{x}^{f+\Delta f}$, $p_{y}^{f+\Delta f}$, and $p_{z}^{f+\Delta f}$ are the 3D coordinates of the joint in the f+Δf frame, Δf is the time step, and Δt is as follows:(7)$$\Delta t=\frac{\Delta f}{FPS}$$
where FPS is the frame speed of the employed Kinect camera. By mapping $v_{x},v_{y},v_{z}$ to R, G, and B, respectively, the joint motion information can be encoded as JMSM with dimensions of $N\times \left(F-\Delta f\right)$.

The proposed method does not impose constraints on the speed value, and takes into account the sudden change in joint position caused by sensor error within the collected data. This is due to the fact that each joint has a different motion range, and imposing constraints on the speed would result in loss of dynamic features for joints with large ranges of motion and fast speeds. Furthermore, analysis of position mutation reveals that joint velocity in the 3D direction is non-zero, and the motion of a joint in an actual scene is interconnected with adjacent joints. Consequently, joints exhibiting position mutation appear as single-color blocks in JMSM. Leveraging this observation, morphological operators can be utilized to enhance texture information in the motion feature map and eliminate noise associated with position mutation joints to improve speed estimation performance. Further details regarding implementation will be provided in the next section.

### 2.4. Image Enhancement

The involvement of each joint varies throughout the entire action sequence. From a visual perspective, the joints with more prominent movement are more likely to capture attention. Based on this observation, the spatial information of joints in SSTM with distinct motion characteristics can be enhanced by leveraging motion energy.

The instantaneous energy possessed by the joint i with coordinates $P_{i}^{\prime f}=(p_{x},p_{y},p_{z})$ in the fth frame within the kth class action sequence can be illustrated as follows:(8)$$\phi_{i}^{f,k}=\left\|P_{i}^{\prime f,k}-P_{i}^{\prime f-1,k}\right\|$$
where f>1, $\left\|\cdot \right\|$ denotes the Euclidean distance. Consequently, the motion energy of the joint i within the whole action sequence can be expressed as follows:(9)$$\Phi_{i}^{k}=\sum_{f=2}^{F}\phi_{i}^{f,k}$$

With the motion energy $\Phi_{i}^{k}$, the color weight $\Omega_{i}^{k}$ of the ith joint can be obtained by the following equation:(10)$$\Omega_{i}^{k}=\frac{\Phi_{i}^{k}-\Phi_{\min}^{k}}{\Phi_{\max}^{k}-\Phi_{\min}^{k}}$$
where $\Phi_{\max}^{k}$ and $\Phi_{\min}^{k}$ are the maximum and minimum values of the motion energy of all joints within the kth action sequence, respectively.

Correspondingly, the enhancement weights relevant to the kth class action motion can be expressed as $W_{k}=\left[\Omega_{25},\cdots ,\Omega_{1}\right]$ according to the above encoding order. Therefore, the enhanced SSTM image can be depicted as follows:(11)$$M_{k}=\left[R_{1}\times W_{k}^{T},R_{2}\times W_{k}^{T},A\times W_{k}^{T}\right]$$

It can be seen from Figure 6a that the colors corresponding to the joint information with high motion energy are enhanced, while the other joint colors are defocused. Consequently, the adaptive enhancement method developed can effectively enhance SSTM motion saliency and, subsequently, improve motion classification performance.

Depth sensors, such as Xtion PRO, introduce noise during the acquisition of joint coordinates, leading to a significant estimation error in the joint information. This hampers recognition capabilities. Focusing on this, the texture information relevant to motion feature map can be enhanced by exploiting morphological operators to improve speed estimation performance. A corrosion operation, which is commonly used to eliminate smaller and meaningless objects, is first performed on the JMSM to eliminate the noise, i.e.,
(12)$$A\Theta E=\left\{\left.z\right|B(z)\right\}\subset A$$
where A is the binary image, B(z) is the result obtained via the corrosion operation Θ, and E is the structure element, which yields
(13)$$I_{v}=\left[v_{x}\Theta E\;v_{y}\Theta E\;v_{z}\Theta E\right]$$

Since the corrosion could change the size of the original image and distort the original texture, the corroded image can be expanded to restore and smooth the original texture, thereby effectively reducing intra-class velocity differences. Consequently, incorporating an expansion operation enables one to achieve
(14)$$I_{v}=\left[\left(v_{x}\Theta E\right)⊕E\;\left(v_{y}\Theta E\right)⊕E\;\left(v_{z}\Theta E\right)⊕E\right]$$
in which ⊕ stands for the expansion operator.

It is shown in Figure 6b that the texture of the enhanced image (second row) is smoother compared to the original image (first row), and the irrelevant information is effectively eliminated while largely preserving the original texture, thereby reducing intra-class action differences.

### 2.5. TS-CNN Design for Action Recognition

Thanks to the unique advantages of feature extraction and representation, CNN is widely used in image recognition, speech processing, and other fields [17,18]. However, traditional CNN-based action recognition methods only utilize a single type of skeleton data, paying less attention to different types of status information, which limits their recognition performance. In contrast to conventional deep networks, TS-CNN can accept multiple types of information as inputs, extract the corresponding deep features separately, and then fuse these extracted features for classification purposes. In light of this information, TS-CNN is exploited here to deeply fuse the extracted static and dynamic skeleton features to improve the action recognition performance.

TS-CNN is exploited via the proposed method to extract and fuse the depth features from SSTM M and JMSM $I_{v}$, to make full use of the space–time and dynamic information of skeleton sequences. The TS-CNN model used consists of two improved AlexNet [19]. SSTM and JMSM are used as the inputs of dynamic and static flow, respectively. Following processing through convolutional layers, pooling layers, and fully connected layers, the posterior probabilities generated by each stream CNN are integrated to yield the final identification result.

The two channels in the constructed TS-CNN have identical structures (shown in Figure 7), ensuring equal weight dimensions and independent updates for each channel. An 8-layer CNN model is chosen, comprising five convolutional layers and three fully connected layers. The first convolutional layer has a convolutional kernel size of 11×11 and a stride of 4. The second layer has a convolutional kernel size of 5×5 with a stride of 2. The next three layers have a convolutional kernel size of 3×3 with a stride of 1. Each convolutional layer uses ReLU activation function to improve the nonlinear mapping and accelerate convergence speed. The first, second, and fifth layers are followed by a maximum pooling layer of size 2 to diminish redundant information and network complexity. Following the fifth convolutional layer, three fully connected layers are employed to comprehensively integrate the extracted depth features. The first and second fully connected layers contain 4096 neurons and dropout can be set to 0.5. For the purpose of preventing network convergence slowdown caused by data distribution change, each convolutional layer is followed by a batch normalization (BN) layer.

Given a skeletal sequence $S_{m},m=1,\cdots ,M$, the SSTM and JMSM are obtained via the above procedure, and then they are scaled to 227×227 pixels by bilinear interpolation for subsequent deep feature extraction. The depth features extracted by using the CNN are outputted to the fully connected layer, which is then normalized by the softmax function to obtain the following posterior probability:(15)$$p_{x}(n\left|x^{m}\right.)=\frac{e^{z_{x}^{n}}}{\sum_{k=1}^{N}e^{z_{x}^{k}}}$$
in which $p_{x}(n\left|x^{m}\right.)$ is the probability of $x^{m}$ belonging to the nth action class, $z_{x}^{k}$ is the output of the fully connected layer corresponding to the kth action class, x stands for SSTM or JMSM, and N is the number of action classes.

The proposed model outputs $p_{SSTM}(n\left|x_{SSTM}^{m}\right.)$ and $p_{JMSM}(n\left|x_{JMSM}^{m}\right.)$ and the stream output are fused by using multiplication fusion to acquire the following final result:(16)$$ActionClass=\arg\max P\left(lab\right)$$
where $P\left(lab\right)=p_{SSTM}(lab\left|x_{SSTM}^{m}\right.)⊙p_{JMSM}(lab\left|x_{JMSM}^{m}\right.)$, 1≤lab≤n, and ⊙ is the dot product operator.

The model parameter can be updated via softmax classifier based on the following loss function:(17)$$L\left(W\right)=-\left[\sum_{m=1}^{M}y^{\left(m\right)}\log(p_{x}(ActionClass^{\left(m\right)}\left|x^{m},W\right.))\right]$$
where W is the model parameter, and $y^{\left(m\right)}$ is the corresponding true label value.

## 3. Experiments and Discussion

The effectiveness of the proposed approach was validated through comparing it with action recognition algorithms based on manually extracted features, RNN models, and CNN models in terms of viewpoint change, subject diversification, and similar forms of action diversification, using the following three publicly available action recognition datasets: NTU RGB-D, Northwestern-UCLA, and UTD MHAD.

### 3.1. Experimental Environment Configuration

For the experimental hardware environment, the mode is built using an Intel Core^(TM)^ i7-7700 processor with 3.60 GHz, 32 GB memory, a NVIDIA GeForce GTX 1070 GPU, and the PyTorch framework with Python 3.7 and CUDA 10.0. Stochastic gradient descent (SGD) is employed to update the network weights, the network learning rate is set to 0.001, momentum is 0.5, and weight decay is 0.00005. The training period is 200, during which 10% of the training set is randomly selected to adjust the training parameters. Additionally, data augmentation techniques such as random vertical flipping, panning, and scaling are employed to increase the number of training samples and enhance the model’s generalization capability.

### 3.2. NTU RGB-D Dataset

The NTU RGB-D dataset comprises action sequences captured simultaneously by three Kinect V2 cameras located at three different views at NTU [9]. The dataset consists of 60 types of actions (including some similar actions, such as reading and writing, etc.) performed by 40 subjects, generating a total of 56,880 skeletal sequences with richly performed subjects and a variety of similar actions with noise (shown in Figure 8, where the third row shows the diversity of similar actions). Currently, this dataset stands as the largest and most challenging action recognition dataset available.

According to the crossover experimental method in [9], the crossover subjects and view experiments are conducted. Specifically, the 40 classes of subjects are divided into training and test sets, where the training sets are numbered as follows: 1, 2, 4, 5, 8, 9, 13, 14, 15, 16, 17, 18, 19, 25, 27, 28, 31, 34, 35, 38. The remaining classes form the test and validation sets, which possesses equal shares of the data. The training along with test and validation sets contain 40,320 and 16,560 samples, respectively.The cross-view trial uses the samples obtained from the first camera to test while all other cameras’ samples serve as training data. The training along with test and validation sets contain 37,920 and 18,960 samples, respectively. 

Figure 9 shows the overall recognition rate of the cross-view experiment conducted by the proposed algorithm on the NTU RGB-D dataset. In Figure 9, each row is the actual category of the action, while each column is the recognition result of the corresponding action achieved by the proposed algorithm, and the main diagonal element indicates the accuracy of action recognition. It can be seen from the confusion matrix shown in Figure 9 that most actions exhibit high recognition rates due to the construction of skeletal spatio-temporal descriptors using relevant inter-joint information and joint dynamic information to jointly characterize skeletal motion features. Furthermore, leveraging motion enhancement and visual enhancement through the developed approach leads to significant improvements in SSTM and JMSM, resulting in a 9.75% increase in recognition rate for certain actions (e.g., sitting down, taking off a jacket) compared to the overall recognition rate. It is worth noting that confusion only arises among actions with subtle differences, such as reading and writing. These findings demonstrate that the proposed method performs well in complex scenes characterized by changing viewpoints, rich noise, and subtle action variations.

The recognition rates of the existing state-of-the-art methods are compared in Table 1. 

Due to the large number of training samples in this dataset, methods based on RNN [9] and LSTM [23] exhibit relatively high recognition rates. Moreover, the proposed method achieved higher accuracy in the cross-view experiment compared to Deep-RNN [9] and ST-LSTM [23], with improvements of 22.78% and 13.87%, respectively, owing to CNN’s effective learning of skeletal space information. Additionally, by incorporating human structure constraints along with joint dynamic information and motion saliency features, the proposed method outperformed TSRJI [16] by 5.77% in terms of recognition rate in the cross-subject experiment. It is worth noting that the recognition rate of the proposed method is 0.97% inferior to that of MM-Net [27], which can be attributed to the fact that MM-Net extracts multi-dimensional features, such as joint distance (JD) and JD velocity (JDV), joint angle (JA) and JA velocity, along with fast-action joint position (FJP) and slow-action joint position (SJP), while the proposed method only acquires space–time and dynamic features, such as SSTM and JMSM.

To assess the robustness of the developed approach with regard to the input noise, an investigation was conducted on this dataset by introducing zero mean input noise subject to a Gaussian distribution with standard deviation σ to the skeleton sequences. The results are presented in Table 2. It is obvious from Table 2 that the proposed method still maintained a high accuracy even with the addition of significant noise (the standard deviation σ was set to 12 cm, which is substantial in the context of the human body’s scale), while MM-Net exhibited significant degradation with the increase in noise level. This can be attributed to the SSTM and JMSM being enhanced via motion and visual enhancement, respectively, to reduce the impact of noise on the accuracy obtained through our method, whereas MM-Net ignores the influence of noise. This demonstrates that our method exhibits considerable robustness against input noise compared to the comparative approach.

### 3.3. Northwestern-UCLA Dataset

The Northwestern-UCLA dataset [28] comprises 1494 sequences featuring 10 action categories captured from diverse viewpoints: picking up with one hand, picking up with both hands, throwing trash, walking, sitting, standing up, putting on, taking off, throwing, and taking. Following the same protocol as in [28], samples obtained from the first two cameras were utilized for training purposes, while the remaining samples served as test and validation data, each of which had 50% of the data.

As shown in Table 3, the recognition rates achieved by deep learning approaches are significantly higher than those obtained through manual feature extraction methods. Secondly, under the assumption that the skeleton is perpendicular to the ground, HOJ3D [29] with skeletal information neglects inter-skeleton relationships and, consequently, exhibits a lower accuracy rate. Moreover, LARP [6], based on associated skeletons with variable parameters, outperforms HOJ3D but fails to consider the temporal information of skeleton sequences. Furthermore, HBRNN-L [8] models inter-skeleton dynamic information to achieve a recognition rate of 78.52%, yet its performance improvement is limited due to relatively small training samples. In contrast, our proposed method encodes spatio-temporal information on skeletons and enhances motion saliency through motion enhancement techniques, resulting in an overall recognition rate improvement of 13.17% compared to LARP. Moreover, by fusing spatial–temporal and motion depth information based on TS-CNN, our proposed method elevates the recognition rate by 8.82% compared to HBRNN-L. Consequently, the developed algorithm demonstrates high action recognition rates for the aforementioned actions, even under viewpoint changes.

### 3.4. UTD MHAD Dataset

UTD-MHAD [30] is a multimodal dataset, comprising data collected from Kinect cameras and wearable inertial sensors. It encompasses 27 classes of actions, with a total of 864 sequences performed by eight test subjects (equally distributed between genders) for each action. In this study, the cross-subject protocol shown in [30] was employed: the odd-numbered tester samples were used for training and the remaining samples were used for testing and validating; each of the sample types had half of the data.

As shown in Table 4, the recognition rate obtained by the proposed method is significantly higher than that of the compared algorithms, which can be attributed to the fact that there are more similar actions in the dataset, such as drawing circles clockwise and counterclockwise. Furthermore, the information associated with the joint time of these actions and the relations among them plays a crucial role in achieving improved accuracy rates, and the above information is integrated in the proposed approach through SSTM and enhanced by exploiting motion energy, thereby enabling higher accuracy rates to be attained.

### 3.5. Ablation Experiments

The effectiveness of each module is validated in this subsection based on NTU RGB-D and UTS-MHAD datasets from the following three viewpoints: coordinate system conversion, feature descriptor construction, and image enhancement.

#### 3.5.1. Effectiveness of Coordinate System Conversion

The coordinate system conversion was conducted on the premise that the correlation between the original joint pairs remains unchanged. Taking the ‘sit down’ action of the eighth type in the NTU RGB-D dataset as an example, the correlation among the first-level-related joint pairs was calculated. It is obvious from Figure 10 that the proposed method maintains the original correlation among joint pairs while enhancing the correlation among some joint pairs. In contrast, the joint pair correlation acquired in [9] is notably lower. It can be seen from the calculation of the correlation coefficient between the joint pair coordinates that the average correlation of the joint pair is 0.89 after transforming the coordinate system by using the proposed approach, which is 0.18 higher than that achieved in [9].

#### 3.5.2. Verification of the Complementarity between SSTM and JMSM

The two feature descriptors, SSTM and JMSM, are designed to capture spatial–temporal and motion information, respectively. After training TS-CNN, the corresponding classification results are presented in Table 5. It can be observed from Table 5 that the fusion of these two feature descriptors leads to an average increase in recognition rate of 8.975% under the NTU RGB-D dataset, indicating a high level of complementarity between them.

#### 3.5.3. Effectiveness of Image Enhancement

The results obtained from comparing the skeletal sequences with and without motion enhancement and visual enhancement on the NTU RGB-D and UTD-MHAD datasets are presented in Table 6. As depicted in Table 6, the recognition rates of the SSTM and JMSM on the UTD-MHAD dataset witnessed improvements of 3.52% and 8.37%, respectively, after motion and visual enhancements were applied. This suggests that the proposed approach for enhancing action recognition is effective in reducing intra-class variation, thereby significantly boosting the overall recognition rate.

## 4. Conclusions

The present study proposed an action recognition method that utilizes 3D skeletal spatio-temporal and dynamic information. Initially, a skeletal spatio-temporal map was constructed using the proposed approach, incorporating inter-joint correlation information while considering the structural constraints of the human body to enhance differentiation among various actions. Subsequently, a joint motion velocity map was established by exploiting the dynamic characteristics of joints. Moreover, motion enhancement was applied to the skeletal spatio-temporal map based on joint motion energy, and morphological operators were employed on the joint motion velocity map for visual enhancement. Finally, TS-CNN was utilized to extract depth features from both the skeletal spatio-temporal map and the joint motion velocity. The classification results obtained from each single stream were fused using multiplicative methods to achieve an overall recognition rate. The experimental evaluations conducted on three public action recognition datasets (NTU RGB-D, Northwestern-UCLA, and UTD MHAD) demonstrated that compared to state-of-the-art methods of action recognition, our proposed approach achieves higher recognition rates in complex scenarios involving viewpoint changes, rich noise levels, diverse subjects, and similar actions.

The developed skeleton spatial–temporal and dynamic-features-based TS-CNN model has several shortcomings that need to be addressed. Firstly, upon analyzing the NTU RGB-D dataset, it was observed that the model easily confuses some actions when only 3D skeleton coordinates are utilized. Considering this issue, we intend to incorporate RGB image features as Supplementary Information to enhance the network’s recognition capabilities. Moreover, upon examining the input feature matrices, it was noted that certain actions exhibit minimal variation between frames, and most of the elements in the matrices remain unchanged. In future work, we will try to incorporate the attention mechanism to enable the network to focus on more valuable information. Furthermore, the constructed model only considers a few features, thereby limiting its performance in action recognition. Future research will explore more global and local features to enhance the recognition performance. Finally, the built model has more parameters compared to the single-stream 2D CNN network, and it therefore exhibits high computational complexity. In the future, we will focus on the model’s optimization to effectively reduce the model’s parameters, and conduct further analyses to determine the most suitable network structure for different features and body parts.

## Figures and Tables

**Figure 1 sensors-24-07609-f001:**
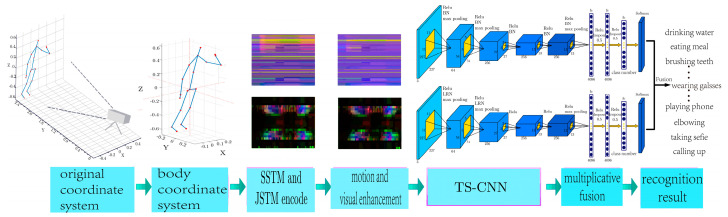
TS-CNN action recognition method with the fusion of skeletal spatial-temporal and motion information.

**Figure 2 sensors-24-07609-f002:**
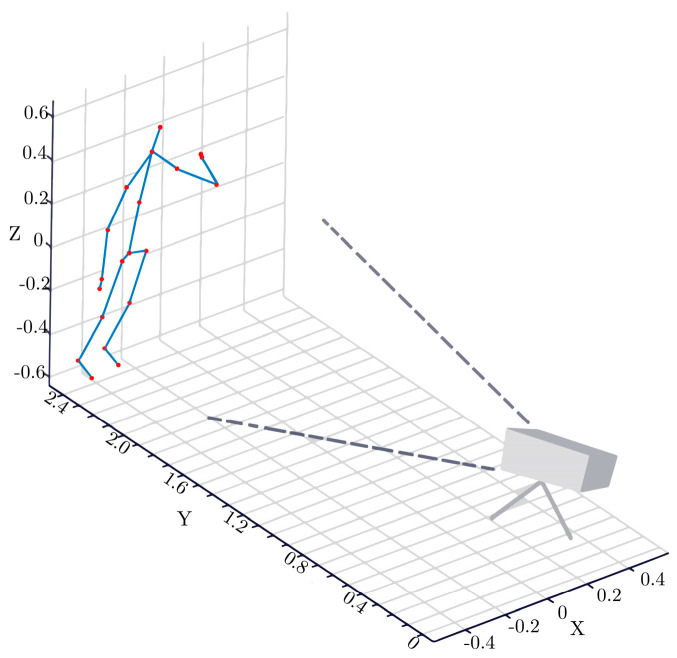
Skeletal coordinates of Kinect coordinate system.

**Figure 3 sensors-24-07609-f003:**
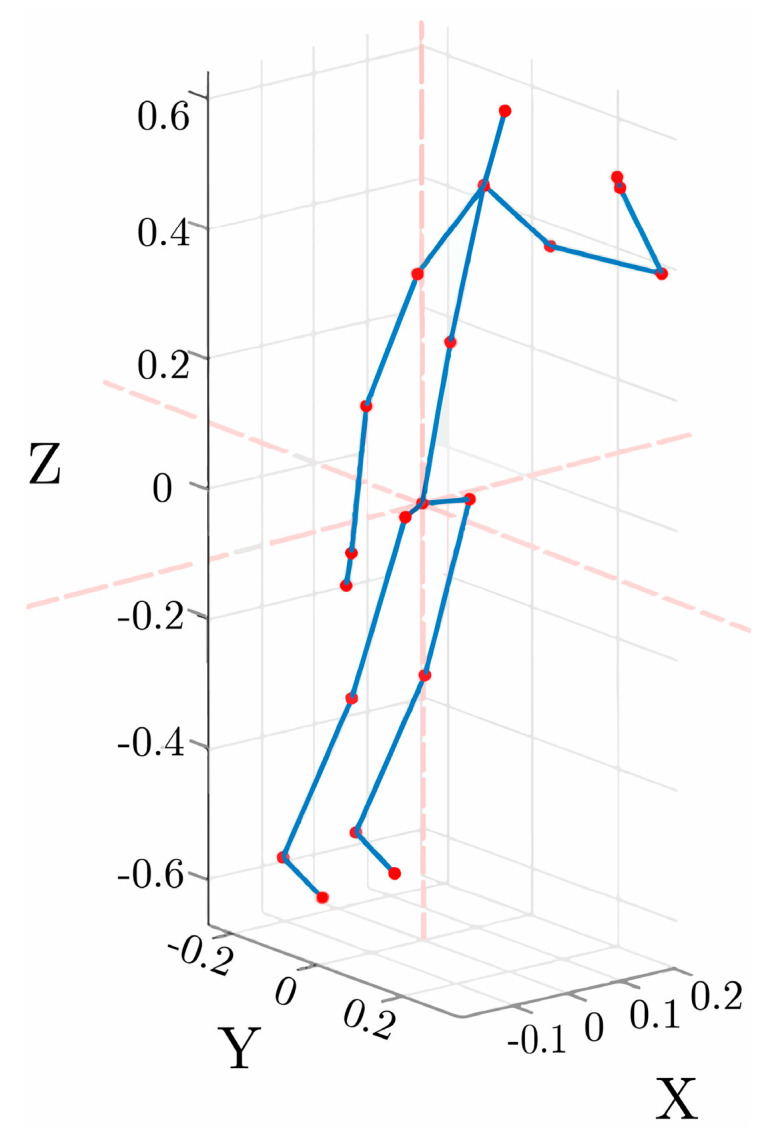
Joint visualization in body coordinate system.

**Figure 4 sensors-24-07609-f004:**
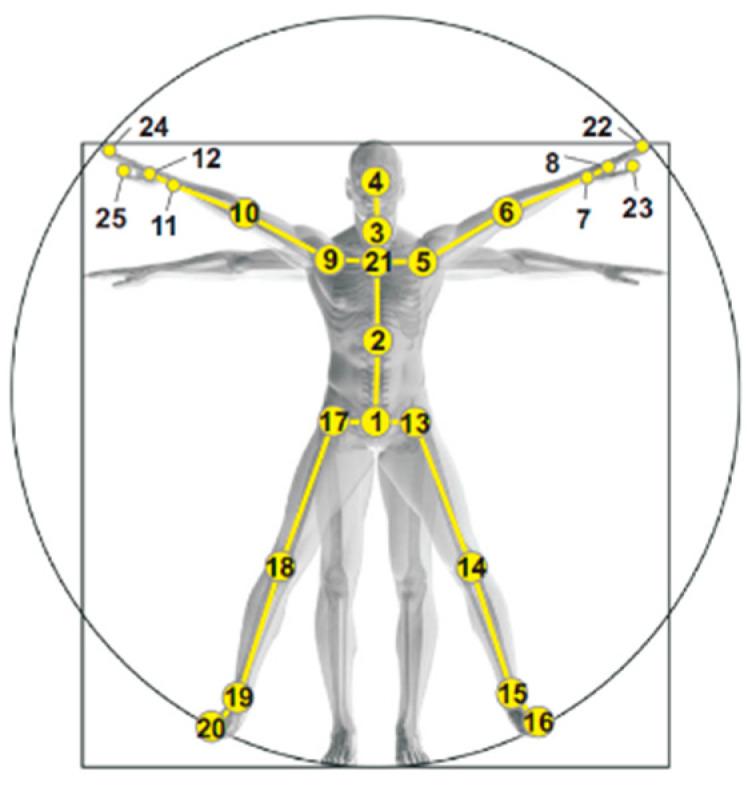
Labeling of 25 human joints.

**Figure 5 sensors-24-07609-f005:**
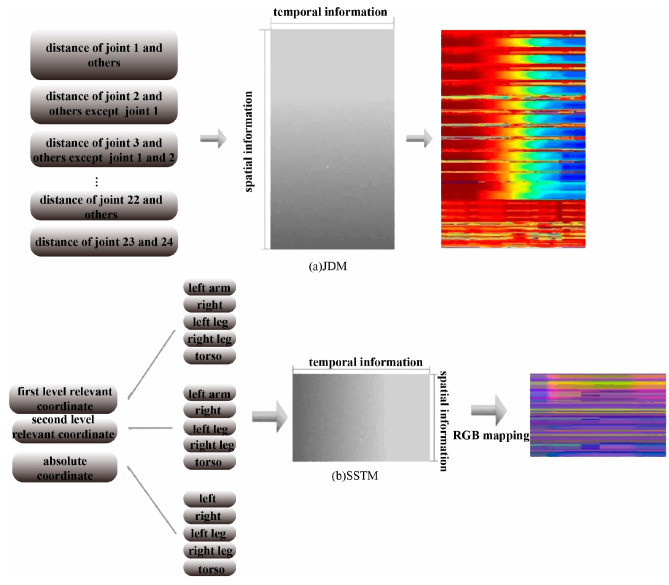
JDM and the proposed SSTM.

**Figure 6 sensors-24-07609-f006:**
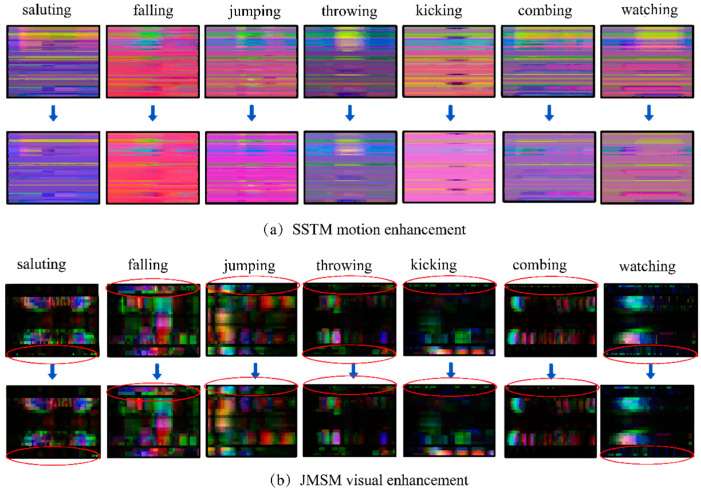
Image enhancement color texture map.

**Figure 7 sensors-24-07609-f007:**
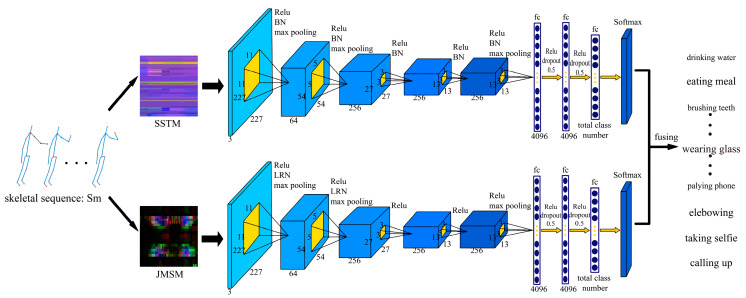
Dual-stream convolutional neural network model.

**Figure 8 sensors-24-07609-f008:**
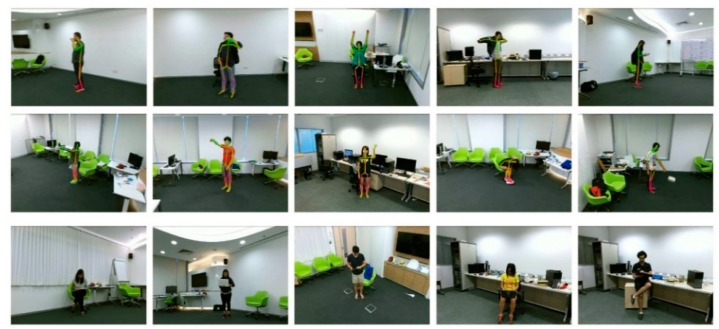
Sample of NTU RGB-D dataset.

**Figure 9 sensors-24-07609-f009:**
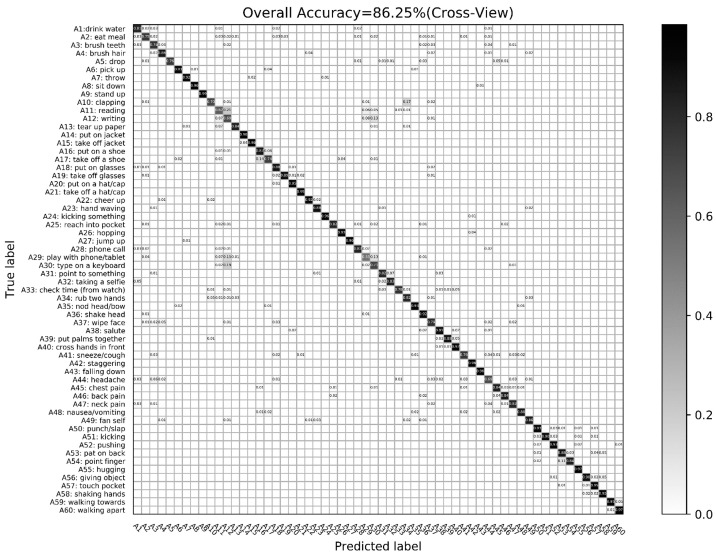
Cross-view verification confusion matrix based on NTU RGB-D dataset.

**Figure 10 sensors-24-07609-f010:**
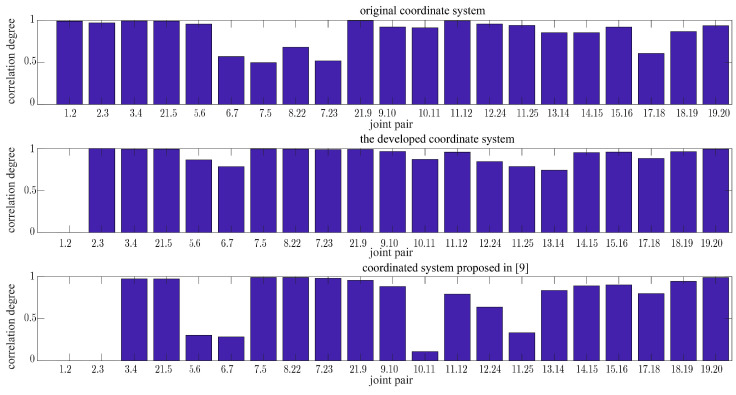
Comparison of joint pair correlations.

**Table 1 sensors-24-07609-t001:** Comparison of experimental results for NTU RGB-D dataset.

Data	Feature	Method	Crossed Subjects	Crossover View
Depth map	Manual extraction	HON4D [20]	30.56%	7.26%
		SNV [21]	31.82%	13.61%
Mixed data	Manual extraction	HOG [22]	32.24%	22.27%
Skeletal sequences	Manual extraction	LARP [6]	50.08%	52.76%
		Dynamic skeletons [7]	60.23%	65.22%
	RNN	HBRNN-L [8]	59.03%	63.97%
		Deep RNN [9]	56.29%	67.29%
		Prae-aware LSTM [9]	62.93%	70.27%
		ST-LSTM [23]	65.20%	76.10%
	CNN	Multi-temporal 3D CNN [24]	66.85%	72.58%
		D-CNN [25]	73.40%	80.40%
		TSRJI [16]	73.30%	80.30%
		Multi-CNN [26]	76.10%	82.64%
		JDM [14]	76.20%	82.30%
		MM-Net [27]	80%	88%
		The developed method	**79.07%**	**86.25%**

**Table 2 sensors-24-07609-t002:** Assessment of robustness against the input noise.

	0.1 cm	1 cm	4 cm	8 cm	16 cm	32 cm
MM-Net	87%	82%	78%	70%	62%	51%
The developed approach	86.1%	84%	80%	75%	68%	58%

**Table 3 sensors-24-07609-t003:** Experimental results for NORTHWESTERN-UCLA dataset.

Data Type	Feature	Method	Accuracy
Depth information	Manual features	HON4D [20]	39.90%
		SNV [21]	42.80%
Skeletal information	Manual features	HOJ3D [29]	54.50%
		LARP [6]	74.20%
	RNN	HBRNN-L [8]	78.52%
	CNN	The proposed method	87.37%

**Table 4 sensors-24-07609-t004:** Experimental results for UTD MHAD dataset.

Method	Accuracy
Kinect & Inertial [30]	79.10%
SOS [31]	86.91%
COv3DJ [32]	85.58%
JTM [33]	85.81%
JDM [14]	88.10%
The developed method	**93.75%**

**Table 5 sensors-24-07609-t005:** Validation of effectiveness of SSTM and JMSM.

Feature Descriptors	Dataset	
	NTU RGB-D (Cross-Subject)	UTD-MHAD
SSTM	67.54%	86.71%
JMSM	72.65%	87.67%
SSTM + JMSM	79.07%	93.75%

**Table 6 sensors-24-07609-t006:** Effectiveness of motion and visual enhancement.

Feature Descriptor	Dataset	
	NTU RGB-D (Cross-Subject)	UTD-MHAD
SSTM	64.21%	84.19%
SSTM + Motion Enhancement	67.54%	87.71%
JMSM	66.44%	79.30%
JMSM + Visual Enhancement	**72.65%**	**87.67%**

## Data Availability

The original data presented in the study are openly available in NTU RGB + D 120 at https://rose1.ntu.edu.sg/dataset/actionRecognition/, accessed on 6 June 2019, UTD-MHAD at https://personal.utdallas.edu/~kehtar/UTD-MHAD.html, accessed on 22 December 2014, and Northwestern-UCLA at http://users.eecs.northwestern.edu/~jwa368/data/, accessed on 2 June 2014.

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
