# Peer review of "Deep Fusion of Skeleton Spatial–Temporal and Dynamic Information for Action Recognition"

_sensors, 2024, doi:10.3390/s24237609_

Round 1
Reviewer 1 Report
Comments and Suggestions for Authors
Aiming at the problem of low recognition rates possessed by traditional deep information-based action recognition algorithms, a spatial-temporal and dynamic features-based action recognition approach is developed, which is conducted on a two-stream convolutional neural network. The skeleton's three-dimensional coordinate system is firstly transformed to acquire relative joint positions and then this relevant joint information is encoded to construct the spatial-temporal feature descriptor, followed by speed information of each joint encoded to obtain the skeleton motion feature descriptor. Moreover, the resultant spatial-temporal and dynamic features can be further enhanced via motion saliency and morphology operators to improve their expression ability. Finally, these enhanced skeleton features are deeply fused via TS-CNN for action recognition implementation. The proposed method is interesting, and has rather high novelty. However, there are some issues to be tackled, which can be listed as follows.
1. Fig. 1 is a bit blurry, please replace it.
2. In eq. (6), the notation |*|should be explained.
3. Please give the explanation of B(z) in eq. (12).
4. What is the mean of the notation ⊕ in eq. (14).
5. There are some expressions should be revised, for example, in line 356, “(TM)” should be superscript; some expression should be simplified, for instance, in line 309, “so as to” should be replaced via “to”, some of the phrase “in order to” in line 61, 109, 172, 327 should be should be replaced with synonyms.
With the description above, this manuscript should be accepted after minor revision.
Comments on the Quality of English Language
The overall quality of the English language in this paper is quite good and largely meets the standards required for publication in this journal. However, there are a few minor grammatical issues and small errors that need to be addressed. I recommend a minor revision to correct these issues.
Author Response
Please see the comments and point-to-point reply in the attached file named Reply to Reviewer1.

Reviewer 2 Report
Comments and Suggestions for Authors
Minors:
Abstract - Please add results (obtained values) to abstract
l.42 " Kinect [5] and Xtion PRO have gained popularity. "
Kinect is discontinued - please add some recent alternatives to this line
Fig.3 Please use another orientation
Fig.5 A lot of gradients, text is not visible, very small image
Fig.6 Too small image.
Tab.4 SSTM row - please fix alignment
l.206 - please fix order of references to [9,10, 25]
l.284 "corrosion operation" - please fix or explain this morphological operator
This part should be supported by the set of example images, moreover.
l.299-300 not important text with references.
l.344 explanation of operator is required (circle with dot)
section 3.1
What about software for the training ?
What about validation set ?
Majors:
1.Please add some 3D noise - this the main problem of 3D estimation from depth camera and check results. It is very important for the sensitivity analysis.
2.Negative results should be discussed also in this paper
Author Response
Please see the comments and point-to-point reply in the attached file named Reply to Reviewer2.

Reviewer 3 Report
Comments and Suggestions for Authors
The article presents a method for human action recognition using a two-stream convolutional neural network (TS-CNN). The method integrates spatio-temporal skeletal features and dynamic information to achieve higher recognition rates in complex environments. The authors provide an analysis that includes skeleton coordinate system transformation, feature encoding, and motion saliency enhancement.
Positive aspects:
1. The proposed method combines skeletal spatio-temporal features with dynamic information, overcoming the limitations of traditional RGB-based recognition methods. This fusion utilizes depth information, which is less affected by environmental variations such as lighting changes and cluttered backgrounds.
2. Transform the coordinate system to a body-centric model with the hip joint as the origin.
3. The article includes experimental results on several corpora.
Shortcomings:
1. Although the method achieves high accuracy, the complexity of the model may limit its practical application.
2. The method relies heavily on high-quality skeletal data and accurate depth information. In real-world scenarios, the quality of these data can vary significantly, potentially affecting the performance of the model. The article does not fully address how the method would perform under less ideal conditions or with noisy data.
3. The article could benefit from a more thorough discussion of its limitations and potential areas for improvement.
4. A notable shortcoming is that the article predominantly provides examples of action recognition methods that are not necessarily the most recent. In the sections where semantic and contextual methods are discussed, the authors describe relatively old methods. However, more modern and advanced methods have been developed and have achieved significant results. For example, if we consider a widely known and available corpus such as AUTSL (see SOTA in paperwithcode - https://paperswithcode.com/sota/sign-language-recognition-on-autsl), the top performing methods in 2023 include (1) STF+LSTM, (2) SAM-SLR, (3) 3D-DCNN + ST-MGCN, many of which are based on MediaPipe and other advanced STF methods.
Comments on the Quality of English Language
Moderate editing of English language required.
Author Response
Please see the comments and point-to-point reply in the attached file named Reply to Reviewer3.

Round 2
Reviewer 2 Report
Comments and Suggestions for Authors
Ok
Author Response
Please refer to the attached file named 'Reply to Reviewer2' for the reply to Reviewer2.

Reviewer 3 Report
Comments and Suggestions for Authors
The authors have improved the article. However, almost all the figures are of poor quality, and it is impossible to read much of the text on them.
Comments on the Quality of English Language
Minor editing of English language required.
Author Response
Please refer to the attached file named 'Reply to Reviewer3' for the reply to the reviewer 3.
